# Pain in High-Dose-Rate Brachytherapy for Cervical Cancer: A Retrospective Cohort Study

**DOI:** 10.3390/jpm13081187

**Published:** 2023-07-26

**Authors:** Ángel Becerra-Bolaños, Miriam Jiménez-Gil, Mario Federico, Yurena Domínguez-Díaz, Lucía Valencia, Aurelio Rodríguez-Pérez

**Affiliations:** 1Department of Anesthesiology, Intensive Care and Pain Medicine, Hospital Universitario de Gran Canaria Doctor Negrín, 35010 Las Palmas de Gran Canaria, Spain; yurena.dd@gmail.com (Y.D.-D.); ori98es@yahoo.es (L.V.); arodperp@gobiernodecanarias.org (A.R.-P.); 2Department of Medical and Surgical Sciences, Universidad de Las Palmas de Gran Canaria, 35010 Las Palmas de Gran Canaria, Spain; 3Department of Anesthesiology, Complejo Hospitalario Universitario Materno Infantil, 35016 Las Palmas de Gran Canaria, Spain; mimi.jg.lp@gmail.com; 4Radiation Oncology Department, Hospital Universitario de Gran Canaria Doctor Negrín, 35010 Las Palmas de Gran Canaria, Spain; mariofedericos@yahoo.it

**Keywords:** pain, periprocedural management, intravenous analgesia, cancer pain, high-dose-rate brachytherapy, cervical cancer, satisfaction

## Abstract

High-dose-rate brachytherapy (HDR) is part of the main treatment for locally advanced uterine cervical cancer. Our aim was to evaluate the incidence and intensity of pain and patients’ satisfaction during HDR. Risk factors for suffering pain were also analyzed. A retrospective study was carried out by extracting data from patients who had received HDR treatment for five years. Postoperative analgesia had been administered using pre-established analgesic protocols for 48 h. Pain assessment was collected according to a protocol by the acute pain unit. Analgesic assessment was compared according to analgesic protocol administered, number of needles implanted, and type of anesthesia performed during the procedure. From 172 patients treated, data from 247 treatments were analyzed. Pain was considered moderate in 18.2% of the patients, and 43.3% of the patients required at least one analgesic rescue. Patients receiving major opioids reported worse pain control. No differences were found regarding the analgesic management according to the intraprocedural anesthesia used or the patients’ characteristics. The number of inserted needles did not influence the postoperative analgesic assessment. Continuous intravenous infusion of tramadol and metamizole made peri-procedural pain during HDR mild in most cases. Many patients still suffered from moderate pain.

## 1. Introduction

Cervical cancer is the second most common gynecological malignancy worldwide [1]. High-dose-rate brachytherapy (HDR), associated with surgery and chemotherapy, has become the cornerstone of the treatment of locally advanced cervical cancer [2]. To perform HDR, it is necessary to insert an applicator or needles near the tumor and to perform a three-dimensional treatment plan using a computed tomography (CT), magnetic resonance imaging (MRI), or positron emission tomography CT (PET-CT). Using a 3D image, it is possible to accurately delineate tumors and surrounding organs at risk, and then prescribe the dose to the target volume. HDR is performed through inserted needles. Following treatment, the needles are left in place for 24 h, after which the treatment is repeated. The needles can be removed after the second session. HDR allows for the administration of high doses of radiotherapy near the area of the tumor and reduces the incidence of injuries secondary to radiotherapy in surrounding tissue. Moreover, HDR has been shown to optimize the applied dose, as it allows the position of colpostates and the duration of the therapy to be modified. In addition, its application has proven to decrease patient immobilization, morbidity, hospital stays, and health costs [2,3]. Thus, HDR has advantages over low-dose-rate brachytherapy (LDR) and external radiotherapy [4,5]. However, HDR has not been shown to reduce anxiety and discomfort, neither during the insertion of needles and colpostates nor during therapy [6].

Peri-procedural pain during HDR is frequent, but its incidence remains unknown. There are multiple causes of postoperative pain in these patients that should be prevented. However, limited data published to date are based on studies analyzing the incidence and intensity of pain during the needle’s insertion [7,8,9,10,11,12,13,14]. The appearance of postoperative pain is associated with systemic complications and an increase in healthcare costs [15], which may offset the advantages provided by HDR. The presence of pain for 24 h after the insertion of needles has not been studied yet and no studies have been carried out to demonstrate which factors can be controlled to diminish its appearance.

The aim of this study was to evaluate the incidence and intensity of pain during the 24 h after needle insertion for HDR and the satisfaction of the patients regarding postoperative analgesia with the different analgesic protocols used in our center. As a secondary objective, intensity of postoperative pain was evaluated in relation to the intraoperative anesthesia used, the characteristics of patients, and the number of needles inserted for the therapy.

## 2. Materials and Methods

Once the approval of the ethics committee of the Hospital Universitario de Gran Canaria Doctor Negrín was obtained (reference 170128), we carried out a retrospective study. Inclusion criteria were all patients who had undergone HDR for cervical cancer from November 2012 to November 2017. Exclusion criteria were age under 18 years old, cognitive impairment, and cognitive disabilities. Cases in which postoperative data collection had not been fully recorded were also excluded. This manuscript follows the STROBE guidelines [16].

### 2.1. Routine Peri-Procedural Management

The choice between general or neuraxial anesthesia for needle insertion was made based on medical considerations or on the patient’s preference. If neuraxial anesthesia was selected, it was performed in the sitting position using 10 mg of 0.5% bupivacaine associated with 10 mcg fentanyl through interspace L2–L3 or L3–L4 to reach a level of sensitive block at dermatome T7–T10. Cases in which general anesthesia was chosen, total intravenous general anesthesia with propofol in continuous infusion associated with 50 mcg fentanyl was used. Mechanical ventilation of patients submitted to general anesthesia was managed using a laryngeal mask. Regardless of the anesthetic treatment carried out during implantation, all patients were given an intravenous dose of the analgesic drugs they would then receive in continuous infusion for the next 24 h.

Postoperative analgesia was given following standardized analgesic protocols designed by the acute pain unit of our hospital, combining different types of drugs administered by electric pumps of continuous perfusion during the 24 postoperative hours (Table 1). The choice of analgesic protocol administered depended on multiple factors, such as medical history, analytical factors, allergies, and the clinical situation of the patients. However, the clinical situation of the patients could have been affected by tumor stage.

### 2.2. Routine Pain Evaluation

The acute pain unit of our service performs daily protocolized follow-up of the analgesic management of patients. Twenty-four hours after the first treatment and just before the second treatment, a doctor from the acute pain unit visited the patient in the ward to assess the analgesic management. The Visual Analogue Scale (VAS) was used to assess pain intensity using a 10 cm line with two extreme points, representing “no pain”—0—and “maximum pain imaginable”—10. In this medical interview, the physician asks the patient to assess the current pain by placing a mark on that line. Subsequently, a ruler is used to measure the distance in centimeters from the “no pain” point to the mark made by the patient. Based on this scale, postoperative pain was categorized into mild (VAS 1–3), moderate (VAS 4–7), and severe (VAS 8–10).

To evaluate patients’ satisfaction in relation to the analgesia received, the subjective assessment scale (SAS) was used. In this medical interview, the physician asks the patient to describe the satisfaction with the management performed using the following categories: excellent, good, fair, and poor. After carrying out the daily protocolized assessment, the physician in charge of the acute pain unit (not involved in this study) enters the data into the unit’s database.

### 2.3. Study Protocol

While reviewing medical records, the following data were collected: age, weight, height, body mass index (BMI), American Society of Anesthesiologists (ASA) physical status, medical and psychiatric comorbidities, drug allergies, toxic habits, and chronic pain analgesic consumption. We also recorded the anesthesia used during the needle placement, the number of needles implanted, and the analgesia administered throughout and after the procedure. We also collected the following data from the acute pain unit database: the protocol of postoperative analgesia administered, the pain intensity assessment, the number of analgesics required during the 24 h period after needle implantation, the side effects of the analgesic medication, and the overall patient satisfaction regarding the postoperative management.

To assess the analgesic quality based on the number of needles inserted for HDR, the patients were stratified into four groups: group 1 (no needles were placed, the treatment was administered through colpostate and a tandem), group 2 (1–5 needles), group 3 (6–10 needles), and group 4 (more than 10 needles).

### 2.4. Statistical Analysis

Qualitative data were expressed as absolute and relative frequencies. Quantitative variables were expressed as mean + standard deviation (SD) in cases of normal variables and median and interquartile rate (IQR) if their distribution was not adjusted to normality. The Kolmogorov–Smirnov test was used to analyze the normality of the data. The analysis of the results of qualitative variables was performed using the Chi-square test. To compare quantitative variables between two groups, the *t*-test for independent samples was used in cases of variables with normal distribution and the Mann–Whitney U test when the distribution of the variables could not be adjusted to normality. To compare quantitative variables among more than two groups, the one-way analysis of variance (ANOVA) test for independent samples was used in cases of variables with normal distribution and the Kruskal–Wallis test in cases where distribution was not adjusted to normality. A *p*-value < 0.05 was considered statistically significant. Data were analyzed using SPS 24.0 (Statistical Package for Social Sciences, IBM, Armonk, NY, USA).

## 3. Results

During the study period, 172 female patients received 260 HDR treatments. Of these patients, data could not be fully retrieved for 18 patients. After excluding these cases, 154 patients who had received 247 treatments were analyzed. Descriptive analysis of the sample is shown in Table 2. Table 3 shows the characteristics of patients according to the protocol administered in each treatment.

Mean VAS during the first 24 h after needle implantation was 2.56 ± 1.04. Pain was considered mild (VAS 1–3) in 81.8% and moderate (VAS 4–7) in 18.2% of cases. No patient presented a VAS higher than 8. During the first 24 h, extra-analgesic medication had to be administered to the 43.3% of the patients. The frequency of the analgesic protocols prescribed and its relationship with pain assessment during the first 24 postoperative hours (mean VAS, SAS registered, and extra analgesia medication) are shown in Table 4. Patients receiving major opioids in their analgesic protocols reported worse pain control (Figure 1).

Neuraxial anesthesia was selected during the needle placement for most patients. Table 5 shows the intraprocedural anesthetic management, as well as the average VAS, SAS, and extra analgesia medication needed for each one. No statistically significant differences were found between general and spinal anesthesia. Patients diagnosed with psychiatric disorders (depression or anxiety, *n* = 40) showed a mean VAS of 2.40 ± 1.01, similar to that shown by patients without this diagnosis (*n* = 207, VAS 2.59 ± 1.04) (*p* = 0.293). No differences were found in terms of the VAS based on smoking habit (*p* = 0.141), alcohol abuse (*p* = 0.256), or preoperative consumption of chronic analgesics (*p* = 0.077).

The average VAS based on the number of needles inserted for the treatment is represented in Figure 2. Figure 3 shows the SAS of each group based on the number of needles inserted. No statistically significant differences among different groups were found in terms of the VAS (*p* = 0.643) or SAS (*p* = 0.935) based on the number of needles inserted. We also found no differences in terms of the analgesic protocol administered in relation to the number of needles inserted (*p* = 0.508).

In 5 of the 247 treatments analyzed (2%), nausea was a side effect of postoperative analgesia. No mortality or severe complications related to the analgesic management were observed during the first 24 h of treatment. We found no relationship between adverse effects and the type of anesthesia applied for needle placement (*p* = 0.483), the analgesic protocol established (*p* = 0.093), or the number of extra-analgesics required (*p* = 0.055). Nor was there a relationship detected between side effects and postoperative VAS (*p* = 0.893) or the number of needles applied (*p* = 0.791).

## 4. Discussion

This study shows that up to 18% of patients undergoing HDR suffered moderate pain during the first 24 h. In addition, 21.4% of patients assessed the received analgesia as fair or poor. Postoperative pain is a frequent cause of delay in hospital discharge and has been shown to increase postoperative morbidity and mortality [15]. Increased blood pressure and myocardial oxygen consumption leads to an increased risk of myocardial ischemia and global respiratory failure in vulnerable patients. Furthermore, the elevation of cortisol levels increases the risk of postoperative infection and hyperglycemia. Postoperative pain constitutes one of the main risk factors for the development of chronic postoperative pain. In addition, patient dissatisfaction increases costs associated with the treatment [15,17]. Brachytherapy interventions can be painful, both during the insertion of applicators and while applicators are in place [8]. Thus, careful analgesic management is necessary after needle placement for HDR. The dilation of the cervix and the upper third of the vagina during the insertion of needles stimulates parasympathetic S2–S4 and pudendal nerves, causing low back pain. The insertion of applicators in the uterus also stimulates sympathetic T10–L1 nerves, provoking lower abdominal pain, nausea, and vomiting [18].

The placement of needles for HDR treatment was performed under anesthesia for all patients in our center. The presence of an anesthesiologist in the brachytherapy team is common in the USA and Europe, having become a key element that allows for optimal levels of analgesia and comfort during therapy [7,8]. However, analgesic management often has to be carried out by radiotherapy oncologists in countries with inadequate availability of anesthesiologists, limiting the use of an appropriate anesthetic technique in developing countries with a higher prevalence of this type of cancer [18]. In the current study, the use of regional or general anesthesia depended on the clinical characteristics of the patient. Combined spinal and epidural anesthesia has shown a tendency to optimize analgesic results and lead to lower narcotic consumption than epidural anesthesia or local anesthesia alone [19]. A denser block of sacral nerve roots is encountered during spinal anesthesia, so vaginal manipulation may be better tolerated than epidural anesthesia [19]. Despite the fact that almost three out of four patients underwent spinal anesthesia in our study, 21.4% of patients evaluated analgesic management as fair–poor. This means that there may be other variables apart from anesthetic management that influence the high perception of pain. On the other hand, even though the VAS found might seem relatively low, around 20% of patients showed a VAS of 4–7. This proportion is high when compared to the VAS obtained in previous studies [8]. Furthermore, the satisfaction observed in our study was good–excellent in 78.5% of patients. This percentage is lower than the satisfaction rates found in previous studies about pain in HDR [8] or even in more aggressive surgeries, such as transvaginal natural orifice transluminal endoscopic surgery [20]. This suggests that postoperative analgesic management can be optimized to favor an adequate evolution of these oncological patients. One of the potential solutions to reduce postoperative pain in our clinical practice could be to remove needles after each treatment. Keeping the needles inserted for 24 h implies immobilizing patients, increasing not only pain or discomfort but also the eventual appearance of complications such as thrombosis. However, it must be taken into account that in our clinical setting, patients frequently come from different geographical areas, so unifying sessions in one hospital admission facilitates the logistics of treatment.

It was difficult to assess optimal anesthetic management during needle implantation with our study due to the disparity of published treatments [7,8,9,10,11,12,13,14,18,19,21,22,23,24,25,26,27,28,29]. Therefore, needle implantation can be performed under different types of sedation [9,18,21]. Both conscious sedation with midazolam and opioids [9], as well as deep sedation with ketamine and propofol [18], have been shown to provide an adequate level of anesthesia in brachytherapy. Although the VAS of patients with deep sedation appears to be lower [18], the recovery described in patients with conscious sedation was faster [9], which may be important in an outpatient setting and with little supervision by anesthesiologists. Some centers perform this procedure under general anesthesia [22,23,24], ensuring immobility of patients during the placement of needles. Although there are no differences in terms of the dose received by the tumor, the dose received by the surrounding tissue (lower rectum) in patients undergoing sedation is significantly higher compared to patients under general anesthesia [23]. Several regional anesthesia methods have been used in cervical brachytherapy: paracervical block and local instillation of local anesthetics [10,19,25], caudal epidural anesthesia [11], spinal anesthesia [12,13], epidural anesthesia [26,27], and combined spinal and epidural anesthesia [13,19,27]. No differences in analgesic efficacy have been found when different anesthetic methods have been compared, such as topical anesthesia with sedation, paracervical nerve block, conscious sedation, or general anesthesia [28]. However, spinal anesthesia has been shown to improve post-device application pain control, with lower rescue analgesic requirements compared to general intravenous anesthesia [29]. Nonetheless, in a study in which only epidural anesthesia was applied in 73 women, the anesthesiologist chose to use general anesthesia as well in 13 patients (17.8%) during placement of the treatment device [26]. In our study, we found no differences in terms of postoperative pain, analgesic medication requirements, or satisfaction between patients who received general anesthesia or spinal anesthesia.

Diagnosis and treatment of cancer is considered a major stress in life and can cause or aggravate psychological disorders [30]. These psychological or physical characteristics may have influenced the high perception of pain. It has been reported that the prevalence of anxiety and depression in patients suffering from uterine cervical cancer is around 52% and 66%, respectively [31]. This high rate of anxious–depressive disorders may be related to the stigma associated with this disease, as it is caused by sexually transmitted strains of HPV [32], which may lead to feelings of guilt and shame. In addition, the immobility required for HDR treatment increases feelings of anxiety and claustrophobia [19], and worsens pain tolerance. On the other hand, brachytherapy itself can result in pain, anxiety, and distress [33]. Undergoing this therapy brings about acute stress in 30% of patients one week after the end of treatment, and post-traumatic stress disorder (PTSD) three months after the therapy in up to 41% [34]. It should be taken into account that most of the patients included in the study underwent two treatment sessions. In addition, the emotional reaction in the previous therapy was not controlled, and no appropriate medication for potential PTSD was administered. Moreover, we found that only 16.9% of our patients had been diagnosed with depression or anxiety, and that no psychiatric reevaluation had been carried out prior to the second treatment. This lower incidence of diagnosed psychiatric disorders implies that many patients may have suffered from a mental disorder that was undiagnosed and, therefore, untreated. We did not find a statistically significant relationship between anxiety/depression and pain, unlike previous studies [35]. This lack of association in the current study may be due to the fact that specific scales were not used to assess anxiety or that the power of the study was insufficient to find such a relationship.

A worse quality of analgesic management could have been expected in patients in which a greater number of needles was implanted. However, taking into account that the main reason for suffering from pain after the needle insertion is the dilation of the cervix and the upper third of the vagina [18], the number of needles would play no role. We confirmed that there was not a statistically significant relationship regarding the number of needles implanted and the assessment of the VAS or SAS. In addition, patients receiving a protocol based on major opioids (protocol D) in the current study were those who showed a higher VAS and lower SAS, and those requiring additional analgesics. It may be that in this retrospective analysis we ignored other variables that were taken into account when administering strong opioids, and that they were prescribed precisely to patients most susceptible to pain. However, there is considerable evidence to suggest that opioid anesthesia does not reduce postoperative pain [36]. Moreover, postoperative opioid requirements may vary among patients. Therefore, a fixed dosing schedule can be an underdose for some patients with special needs and an overdose for others [37].

We acknowledge some limitations of our study. The main one is its retrospective nature. However, the acute pain unit of our service carried out a strict protocol during the postoperative follow-up, collecting all data related to the pain daily and objectively, and data loss was rare. In addition, there were variables that were not taken into account in this analysis that could have affected the analgesic assessment of patients, such as tumor stage. Since this is a retrospective study in which routine clinical practice carried out over 5 years was evaluated, it was not possible to establish a baseline control group. A prospective analysis could confirm data obtained and allow for the investigation of other variables that were not studied, and a randomized clinical trial could confirm which one is the best analgesic alternative. Second, we did not focus on the evolution of patients after hospital discharge. Since the period under review covered five years, it was difficult to compare the evolution of patients without taking more variables into account. It would have been interesting to evaluate other characteristics, such as the number of hospital readmissions or visits to the emergency department due to pain after hospital discharge, the intensity of pain beyond 24 h after the procedure, the worsening of patients’ psychiatric comorbidities, the incidence of chronic pelvic pain after therapy, cancer-free time, or survival. However, the collection time would be too long and impractical for many of these variables in a study focused on analyzing postoperative acute pain.

## 5. Conclusions

Patients with cervical cancer undergoing HDR experience pain and distress. Postoperative analgesic management using continuous intravenous infusion of combined tramadol and metamizole protocolized and adapted to the clinical characteristics makes postoperative pain mild in most cases. However, many patients still suffer from moderate pain.

Postoperative analgesic management can be optimized to increase satisfaction levels. The number of needles inserted for the therapy does not seem to be a risk factor for worsening acute postoperative pain in HDR. The presence of psychiatric disorders such as anxiety or depression makes it difficult to manage postoperative pain. Therefore, attention to the examination of psychological and psychiatric factors may be more important than the mere administration of strong opioids.

## Figures and Tables

**Figure 1 jpm-13-01187-f001:**
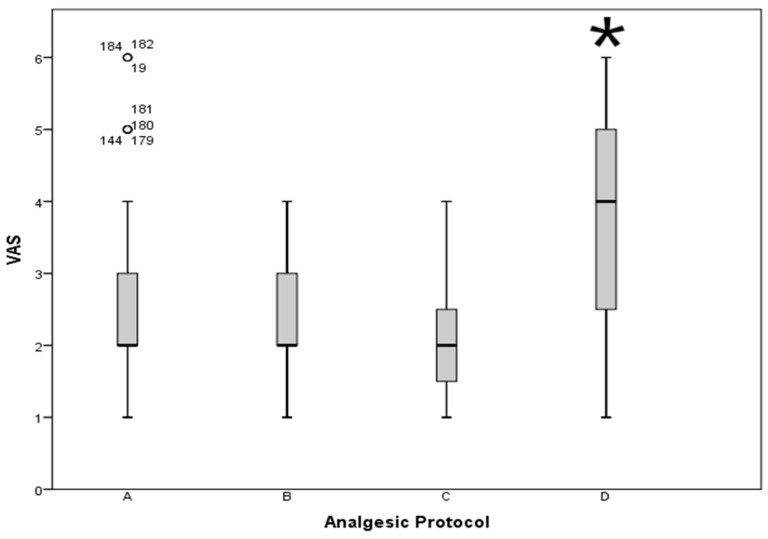
Visual Analogue Scale according to the analgesic protocol. VAS: Visual Analogue Scale; A: tramadol (400–300 mg) + NSAIDs (metamizole 12 g or dexketoprofen 300 mg) + antiemetic (metoclopramide 60 mg); B: tramadol (150 mg) + NSAIDs (metamizole 12 g) + antiemetic (metoclopramide 60 mg); C: NSAIDs (metamizole 12 g or dexketoprofen 300 mg); D: morphine + PCA + metamizole 2 g or acetaminophen 1g every 8 h. * *p* = 0.006 vs. A, *p* = 0.002 vs. B, and *p* = 0.023 vs. C.

**Figure 2 jpm-13-01187-f002:**
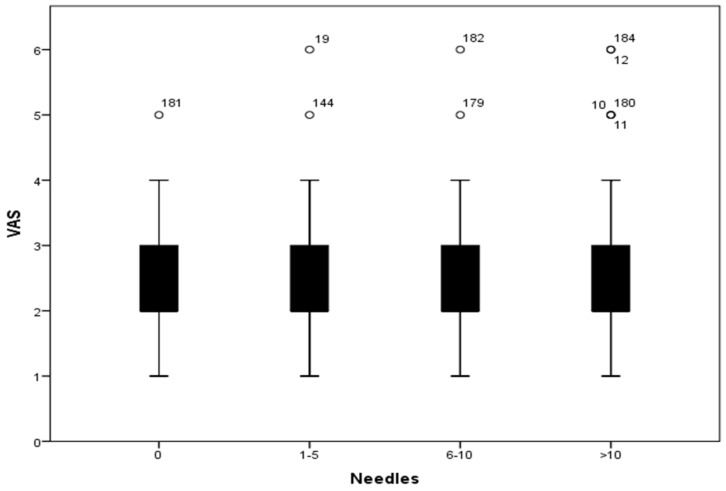
Visual Analogue Scale based on the number of needles inserted (categorized). VAS: Visual Analogue Scale; 0: no needles inserted; 1–5: 1 to 5 needles inserted; 6–10: 6 to 10 needles inserted; >10: more than 10 needles inserted.

**Figure 3 jpm-13-01187-f003:**
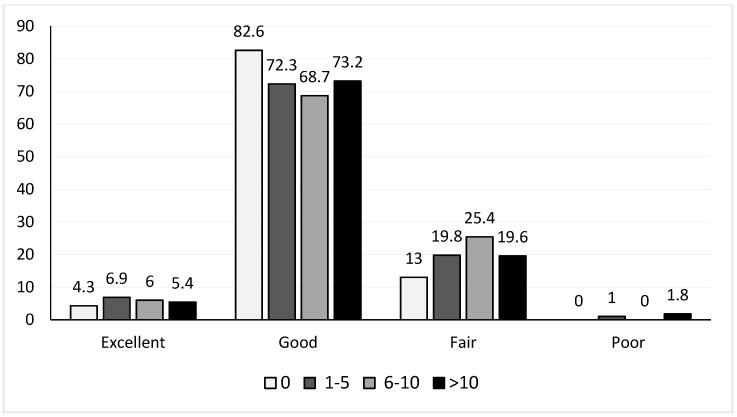
Subjective Assessment Scale based on the number of needles implanted. Data are expressed as relative frequencies (%). SAS: Subjective Assessment Scale; 0: no needles inserted; 1–5: 1 to 5 needles inserted; 6–10: 6 to 10 needles inserted; >10: more than 10 needles inserted.

**Table 1 jpm-13-01187-t001:** Analgesic protocols administered intravenously postoperatively.

Analgesic Protocols	Drugs
Protocol A	Tramadol (400–300 mg) + NSAIDs (metamizole 12 g or dexketoprofen 300 mg) + antiemetic (metoclopramide 60 mg)
Protocol B	Tramadol (150 mg) + NSAIDs (metamizole 12 g) + antiemetic (metoclopramide 60 mg)
Protocol C	NSAIDs (metamizole 12 g or dexketoprofen 300 mg)
Protocol D	Morphine + PCA + metamizole 2 g or acetaminophen 1 g each 8 h

NSAIDs: non-steroidal anti-inflammatory drugs; PCA: patient-controlled analgesia.

**Table 2 jpm-13-01187-t002:** Patient characteristics and pain assessment for each treatment.

Patients (*n* = 154)		
Age, years old		54 (43–63)
Weight, kg		63 (56–70)
BMI, kg·m^−2^		23.9 (21.1–27.4)
ASA physical status, *n* (%)	II	59 (38.3)
	III	95 (61.7)
Smoker, *n* (%)	Current smoker	48 (31.2)
	Ex-smoker	23 (15.0)
Alcohol consumption, *n* (%)	Current	8 (5.2)
	Former	1 (0.7)
Psychiatric comorbidities, *n* (%)	Depression	17 (11.0)
	Anxiety	9 (5.8)
Therapy for chronic pain, *n* (%)		11 (7.1)
**Treatments (*n* = 247)**		
VAS, *n* (%)	1	26 (10.5)
	2	116 (47.0)
	3	60 (24.3)
	4	35 (14.2)
	5	6 (2.4)
	6	4 (1.6)
SAS, *n* (%)	Excellent	15 (6.1)
	Good	179 (72.5)
	Fair	51 (20.6)
	Poor	2 (0.8)
Additional analgesic requirements, *n* (%)	0	140 (56.7)
	1	44 (17.8)
	≥2	63 (25.5)

Data are expressed as frequencies (%) or median (IQR). BMI: body mass index; VAS: Visual Analogue Scale; SAS: Subjective Assessment Scale.

**Table 3 jpm-13-01187-t003:** Patient characteristics and protocol administered in each treatment.

Treatments (*n* = 247)						
		Protocol A (*n* = 187)	Protocol B (*n* = 42)	Protocol C (*n* = 11)	Protocol D (*n* = 7)	*p*-Value
Age, years old		54 (51–56)	60 (55–65)	67 (55–78)	46 (36–57)	0.001
Weight, kg		52 (49–55)	63 (56–69)	66 (55–76)	48 (45–54)	0.001
BMI, kg·m^−2^		23.9 (22.9–24.9)	25.0 (22.2–27.8)	24.6 (21.5–27.8)	20 (19.1–25.4)	0.210
Anesthesia duration, min		82 ± 27	90 ± 36	105 ± 35	113 ± 45	0.055
ASA physical status, *n* (%)	II	78 (41.8)	11 (26.2)	1 (9.1)	7 (100)	0.223
	III	109 (58.2)	31 (73.8)	10 (90.9)	0 (0)	
Smoker, *n* (%)	Current	58 (31.0)	8 (19.0)	3 (27.3)	3 (42.8)	0.921
	Ex-smoker	29 (15.5)	15 (35.7)	2 (18.2)	0 (0)	
Alcohol consumption, *n* (%)	Current	8 (4.3)	0 (0)	1 (9.1)	0 (0)	0.977
	Former	2 (1.1)	0 (0)	0 (0)	0 (0)	
Psychiatric comorbidities, *n* (%)	Depression	29 (15.5)	6 (14.3)	0 (0)	3 (42.8)	0.811
	Anxiety	4 (2.1)	3 (7.1)	3 (27.3)	0 (0)	
Therapy for chronic pain, *n* (%)		8 (4.3)	4 (9.5)	3 (27.3)	2 (28.5)	0.027

Data are expressed as frequency (%), median (IQR), or mean + SD. BMI: body mass index. A: tramadol (400–300 mg) + NSAIDs (metamizole 12 g or dexketoprofen 300 mg) + antiemetic (metoclopramide 60 mg); B: tramadol (150 mg) + NSAIDs (metamizole 12 g) + antiemetic (metoclopramide 60 mg); C: NSAIDs (metamizole 12 g or dexketoprofen 300 mg); D: morphine + PCA + metamizole 2 g or acetaminophen 1 g every 8 h.

**Table 4 jpm-13-01187-t004:** Analgesic protocols administered and postoperative analgesic assessment.

Analgesic Protocol, *n* (%)		VAS	SAS, *n* (%)		Additional Analgesic Requirements, *n* (%)	
Protocol A	187 (73.2)	2.56 ± 1.05	Excellent	13 (6.9)	None	101 (54.0)
			Good	133 (71.1)	Once	34 (18.2)
			Fair	39 (20.9)	Twice	34 (18.2)
			Poor	2 (1.1)	Three-times	18 (9.6)
Protocol B	42 (20.7)	2.48 ± 0.74	Excellent	1 (2.4)	None	25 (59.5)
			Good	36 (85.7)	Once	9 (21.4)
			Fair	5 (11.9)	Twice	6 (14.3)
			Poor	0 (0.0)	Three-times	2 (4.8)
Protocol C	11 (3.3)	2.09 ± 0.94	Excellent	0 (0.0)	None	8 (72.7)
			Good	9 (81.8)	Once	1 (9.1)
			Fair	2 (18.2)	Twice	2 (18.2)
			Poor	0 (0.0)	Three-times	0 (0)
Protocol D	7 (2.9)	3.71 ± 1.79	Excellent	1 (14.3)	None	6 (85.7)
			Good	1 (14.3)	Once	0 (0)
			Fair	5 (71.4)	Twice	1 (14.3)
			Poor	0 (0.0)	Three-times	0 (0)
*p*-value		0.010	0.034		0.659	

Data are expressed as frequency (%) or mean + SD. VAS: Visual Analogue Scale. SAS: Subjective Assessment Scale. A: tramadol (400–300 mg) + NSAIDs (metamizole 12 g or dexketoprofen 300 mg) + antiemetic (metoclopramide 60 mg); B: tramadol (150 mg) + NSAIDs (metamizole 12 g) + antiemetic (metoclopramide 60 mg); C: NSAIDs (metamizole 12 g or dexketoprofen 300 mg); D: morphine + PCA + metamizole 2 g or acetaminophen 1 g every 8 h.

**Table 5 jpm-13-01187-t005:** Intraprocedural anesthesia provided and postoperative analgesia assessment.

Intraprocedural Anesthesia, *n* (%)		VAS	SAS, *n* (%)		Additional Analgesic Requirements, *n* (%)	
General	65 (26.3)	2.36 ± 0.86	Excellent	3 (4.6)	None	39 (60.0)
			Good	52 (80.0)	Once	9 (13.8)
			Fair	10 (15.4)	Twice	14 (21.5)
			Poor	0 (0.0)	Three times	3 (4.6)
Spinal	182 (73.7)	2.62 ± 1.09	Excellent	12 (6.6)	None	101 (55.5)
			Good	127 (69.8)	Once	35 (19.2)
			Fair	41 (22.5)	Twice	29 (15.9)
			Poor	2 (1.1)	Three times	17 (9.3)
*p*-value		0.092	0.408		0.371	

Data are expressed as frequency (%) or mean + SD. VAS: Visual Analogue Scale. SAS: Subjective Assessment Scale.

## Data Availability

Data are available upon reasonable request to the corresponding authors.

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
