# Peer review of "Pain in High-Dose-Rate Brachytherapy for Cervical Cancer: A Retrospective Cohort Study"

_jpm, 2023, doi:10.3390/jpm13081187_

Round 1

Reviewer 1 Report

The article under adjudication is certainly relevant in the context of HDR as a tool for cancer treatment. I have the following comments to the authors:

1) The current method section needs the following elaborations:

a) Inclusion and screening criteria of subjects chosen for the study.

b) how exactly pain was evaluated from the patients need to be mentioned.

c) effectiveness of the cancer treatment/mortalities need to be mentioned in the result.

d) Not sure if there can be a control group assigned for the baseline of this evaluation. Can the authors shed any light on it?

2) Table 3: Can the protocols be connected to the stage of cancer that was corresponding to it? 

3)How is excellent, good, fair etc are categorized - if a rubric of the scale is provided in methods, it would be great for general readership. Fig 3/Table 3 related.

4)In the discussion, it would be great if the authors could shed some light on the type of analgesic that is being used and which would be a better choice over the other. I can see the number of needles directly correlates to pain and it is an obvious outcome. A deeper insight is solicited.

English language in text is fine and legible. 

Author Response

Reviewer #1

Q1) The article under adjudication is certainly relevant in the context of HDR as a tool for cancer treatment. I have the following comments to the authors: 1) The current method section needs the following elaborations: a) Inclusion and screening criteria of subjects chosen for the study.

R1.- Thank you very much for the positive comments regarding the manuscript.

Inclusion criteria were already stated in the original manuscript: Those patients who had undergone HDR for cervical cancer from November 2012 to November 2017 were included” (lines 66 – 67). However, at the request of the Reviewer, we have changed the statement in the revised manuscript, with the intention of making inclusion criteria more explicit. Now it reads: “Inclusion criteria were: all patients who had undergone HDR for cervical cancer from November 2012 to November 2017” (lines 70 –71).

Q2) b) how exactly pain was evaluated from the patients need to be mentioned.

R2.- We truly appreciate this comment. We have added a section in the revised manuscript entitled Routine pain evaluation. Taking into consideration the Reviewer’s view, we have added a more detailed explanation of the analgesic evaluation performed: “Twenty-four hours after the first treatment and just before the second treatment, a doctor from the acute pain unit visits the patient on the ward to assess the analgesic management. The Visual Analogue Scale (VAS) was used to assess pain intensity, using a 10-centimeter line, with two extreme points representing 0 (no pain) and 10 (maximum possible pain). In this medical interview, the physician asks the patient to assess the current pain by placing a mark on that line. Subsequently, a ruler is used to measure the distance in centimeters from the “no pain” point to the mark made by the patient” (lines 96 – 103).

Q3) c) effectiveness of the cancer treatment/mortalities need to be mentioned in the result.

R3.- Thank you very much for the opportunity to clarify this issue. We acknowledge the interest in showing the results in mortality and the effectiveness of cancer treatment. However, it should be taken into account that the main objective of this study is to assess the incidence and intensity of pain during the 24 hours after needles insertion for HDR and the satisfaction of patients with the analgesic management. Showing results of data collected for a duration after these 24 hours is out of the interest of this study. In addition, this retrospective study includes patients treated during 5 years, so the morality may have been influenced by factors beyond those controlled in the present analysis. This limitation is already stated in the Discussion section “… we did not focus on the evolution of patients after hospital discharge. Since the period under review covered five years, it was difficult to compare the evolution of patients without taking more variables into account” (lines 319 – 321).

               However, in response to the Reviewer’s concern regarding mortality, we have added the following sentence in the Results section of the revised manuscript, within the paragraph on complications secondary to treatment: “No mortality or severe complications related to the analgesic management were observed during the first 24 hours of treatment” (lines 198 – 199).

Q4) d) Not sure if there can be a control group assigned for the baseline of this evaluation. Can the authors shed any light on it?

R4.- Thank you for this comment. Since this is a retrospective study in which routine clinical practice carried out over 5 years is evaluated, it was not possible to establish a baseline control group. Furthermore, taking into account that pain is a subjective variable, it is difficult to establish control groups in which all factors are controlled. Because of the importance of this comment, we have added this information in the limitations paragraph of the Discussion section (lines 315 – 316).

Q5) 2) Table 3: Can the protocols be connected to the stage of cancer that was corresponding to it? 

R5.- Thank you very much for this comment. The choice of the analgesic protocol administered depended on multiple factors, such as medical history, analytical factors, allergies, or the clinical situation of patients. However, tumor stage can affect the clinical situation of patients. This information has been added to the Methods section (lines 89 – 91). The anesthesiologist responsible for the analgesic protocol was frequently unaware of the stage of cancer, and therefore did not usually administer analgesics based on it.  

               We have also added the following information to the limitations paragraph: “In addition, there were variables that were not taken into account in this analysis and that could affect the analgesic assessment of patients, such as tumor stage (lines 313 – 315).

Q6) 3)How is excellent, good, fair etc are categorized - if a rubric of the scale is provided in methods, it would be great for general readership. Fig 3/Table 3 related.

R6.- Thank you very much for this suggestion. We have added all the information regarding the analgesic evaluation in the “Routine pain evaluation” subsection. Regarding patient’s satisfaction, we have added the following: “In this medical interview, the physician asked the patient to describe the satisfaction with the management performed, using the following categories: excellent, good, fair or poor” (lines 106 – 108).

Q7) 4)In the discussion, it would be great if the authors could shed some light on the type of analgesic that is being used and which would be a better choice over the other. I can see the number of needles directly correlates to pain and it is an obvious outcome. A deeper insight is solicited.

 R7.- Thank you very much for this comment. We agree that it would have been great to be able to specify what would be the best analgesic alternative. However, it is not possible to conclude causality due to the characteristics of the study. In addition, variability of analgesic management detected in the literature about HDR increases the difficulty in concluding that an analgesic/anesthetic management is more appropriate than others.

In order to state it clearly in the limitation section of the revised manuscript we have added the following: “… and a randomized clinical trial could confirm which is the best analgesic alternative” (lines 318 – 319).

Reviewer 2 Report

This paper discusses an approach to alleviate pain during high-dose-rate brachytherapy (HDR-BT) for cervical cancer. HDR-BT is an indispensable treatment for curative therapy of cervical cancer, although it is accompanied by pain. Pain management is currently performed empirically and lacks standardization across facilities. Therefore, this paper may hold potential usefulness.

1.       The current title (Pain in high-dose-rate brachytherapy for cervical cancer) seems to be that of a review article, so it would be preferable to have a title that specifically reflects the content of this paper.

2.       In the INTRODUCTION session, it is stated that to perform MRI is necessary for HDR. However, globally, CT-based 3D-IGBT is also widely practiced. Consequently, a revision of the description in this section should be considered.

3.       Is it not an alternative approach, deviating from the global standard, to leave the needle in place for over 24 hours and repeat the HDR? Shouldn't the standard practice involve the removal of the needle after each HDR procedure? It is only natural to experience continued discomfort even after 24 hours, as the needle remains inserted. Rather than modifying the analgesic protocol, wouldn't it be more beneficial to consider altering the methodology of HDR for pain management? To make matters worse, keeping the needle in place poses the risk of thrombosis. Please consider to include these aspects into the section of DISCUSSION.

4.       Please spell out ASA as "American Society of Anesthesiologists-Physical Status" in its initial occurrence. As the authors have mentioned, in countries with a high incidence of cervical cancer, it is often the case that anesthesiologists are not involved in providing pain relief for HDR. Considering that this paper is more commonly read by radiation oncologists than anesthesiologists, it would be beneficial to spell out ASA, which is considered common knowledge for anesthesiologists.

5.       As the authors have articulated in their discussion, I believe that the major cause of pain lies in the dilation of the uterus and vagina. Factors such as age (postmenopausal) and physique (narrow vagina) may potentially influence the severity of pain. Therefore, it is crucial to present which patients were assigned to each analgesic protocol, as this information holds significant value. Please consider presenting this information in the paper.

Author Response

Reviewer #2

Q1) This paper discusses an approach to alleviate pain during high-dose-rate brachytherapy (HDR-BT) for cervical cancer. HDR-BT is an indispensable treatment for curative therapy of cervical cancer, although it is accompanied by pain. Pain management is currently performed empirically and lacks standardization across facilities. Therefore, this paper may hold potential usefulness.

R1.- We truly thank the Reviewer for the positive comments regarding the manuscript.

Q2) 1.The current title (Pain in high-dose-rate brachytherapy for cervical cancer) seems to be that of a review article, so it would be preferable to have a title that specifically reflects the content of this paper.

 R2.- Thank you very much for this comment. Following the Reviewer’s suggestion, we have changed the title accordingly (line 2).

Q3) 2. In the INTRODUCTION session, it is stated that to perform MRI is necessary for HDR. However, globally, CT-based 3D-IGBT is also widely practiced. Consequently, a revision of the description in this section should be considered.

 R3.- Thank you for this comment. We completely agree with the reviewer, that not only MRI is useful to plan treatment. Therefore, we have corrected this information in the Introduction section of the revised manuscript: To perform HDR, it is necessary to insert an applicator or needles near the tumor and to perform a three-dimensional treatment plan using a computed tomography (CT), magnetic resonance imaging (MRI), or positron emission tomography-CT (PET-CT). Using a 3D image, it is possible to accurately delineate tumors and surrounding organs at risk, and then prescribe the dose to the target volume (lines 37 – 42).    

Q4) 3. Is it not an alternative approach, deviating from the global standard, to leave the needle in place for over 24 hours and repeat the HDR? Shouldn't the standard practice involve the removal of the needle after each HDR procedure? It is only natural to experience continued discomfort even after 24 hours, as the needle remains inserted. Rather than modifying the analgesic protocol, wouldn't it be more beneficial to consider altering the methodology of HDR for pain management? To make matters worse, keeping the needle in place poses the risk of thrombosis. Please consider to include these aspects into the section of DISCUSSION.

 R4.- Thank you very much for the opportunity to clarify this issue. Perhaps the main problem with analgesic management is that there is no standardized approach in HDR among different hospitals. Following the Reviewer’s suggestion, we have added the following explanation in the Discussion section of the revised manuscript: “One of the potential solutions to reduce postoperative pain in our clinical practice could be to remove needles after each treatment. Keeping the needles inserted for 24 hours implies immobilizing patients, increasing not only pain or discomfort, but also the eventual appearance of other complications such as thrombosis. However, it must be taken into account that in our clinical setting, patients frequently come from different geographical areas, so unifying sessions in one hospital admission facilitates the logistics of treatment” (lines 242 – 248).

Q5) 4. Please spell out ASA as "American Society of Anesthesiologists-Physical Status" in its initial occurrence. As the authors have mentioned, in countries with a high incidence of cervical cancer, it is often the case that anesthesiologists are not involved in providing pain relief for HDR. Considering that this paper is more commonly read by radiation oncologists than anesthesiologists, it would be beneficial to spell out ASA, which is considered common knowledge for anesthesiologists.

R5.- Thank you very much. We have spelled out ASA in the revised manuscript (line 113).  

Q6) 5. As the authors have articulated in their discussion, I believe that the major cause of pain lies in the dilation of the uterus and vagina. Factors such as age (postmenopausal) and physique (narrow vagina) may potentially influence the severity of pain. Therefore, it is crucial to present which patients were assigned to each analgesic protocol, as this information holds significant value. Please consider presenting this information in the paper.

R6.- Thank you for this comment. We completely agree with the Reviewer. It is already stated in the Discussion section that “the dilation of the cervix and the upper third of the vagina during the insertion of needles stimulates parasympathetic S2 – S4 and pudendal nerves, causing low back pain. The insertion of applicators in the uterus also stimulates sympathetic T10 – L1 nerves” (lines 216 – 219).

There are many factors that may influence the severity of pain. As this is a retrospective study, unfortunately, we cannot obtain some data, such as the size of the vagina. However, we do have data on the characteristics of patients according to the protocol administered. Initially, we avoided adding this information as we believed that it could complicate the understanding of the manuscript, since it refers to the protocols administered in each treatment (n = 247). This information differs from that shown in table 2, which refers to the protocol administered to each patient (n = 145). In addition, the statistically significant relationships detected among different groups may be due merely to the differences in sample sizes among protocols, resulting in merely chance findings that can confuse potential readers.

We have added this information in the revised manuscript (Table 3, lines 144 – 145) due to the Reviewer’s request. However, for the above reasons, we also leave the final decision on whether to display it in the final version of the manuscript to the Editor.

Round 2

Reviewer 2 Report

I believe this paper has been appropriately revised and is worthy of publication.